# Addressing a Pre-Clinical Pipeline Gap: Development of the Pediatric Acute Myeloid Leukemia Patient-Derived Xenograft Program at Texas Children’s Hospital at Baylor College of Medicine

**DOI:** 10.3390/biomedicines12020394

**Published:** 2024-02-08

**Authors:** Alexandra M. Stevens, Maci Terrell, Raushan Rashid, Kevin E. Fisher, Andrea N. Marcogliese, Amos Gaikwad, Pulivarthi Rao, Chelsea Vrana, Michael Krueger, Michael Loken, Andrew J. Menssen, Jacqueline A. Cook, Noah Keogh, Michelle Alozie, Hailey Oviedo, Alan K. Gonzalez, Tamilini Ilangovan, Julia Kim, Sohani Sandhu, Michele S. Redell

**Affiliations:** 1Section of Hematology/Oncology, Department of Pediatrics, Texas Children’s Cancer and Hematology Center, Baylor College of Medicine, 1102 Bates St, Suite 750, Houston, TX 77030, USAmlredell@texaschildrens.org (M.S.R.); 2Department of Pathology & Immunology, Baylor College of Medicine, Genomic Medicine Division, Texas Children’s Hospital, Houston, TX 77030, USA; 3Department of Pathology & Immunology, Baylor College of Medicine, Laboratory Medicine Division, Texas Children’s Hospital, Houston, TX 77030, USA; 4Hematologics, Seattle, WA 98121, USA

**Keywords:** PDX development, Children’s Oncology Group (COG), serial transplanting, BCM PDX portal, future development, NSGS, MISTRG, MISTRG6, novel therapeutics

## Abstract

The survival rate of pediatric acute myeloid leukemia (pAML) is currently around 60%. While survival has slowly increased over the past few decades, the development of novel agents likely to further improve survival for this heterogeneous patient population has been limited by gaps in the pAML pre-clinical pipeline. One of the major hurdles in evaluating new agents for pAML is the lack of pAML patient-derived xenograft (PDX) models. Unlike solid tumors and other types of leukemias, AML is notoriously hard to establish in mouse models, likely due in part to the need for specific human microenvironment elements. Our laboratory at TCH/BCM addressed this gap by establishing a systematic PDX workflow, leveraging advanced immunodeficient hosts and capitalizing on our high volume of pAML patients and close coordination between labs and clinical sections. Patients treated at TCH are offered the chance to participate in specimen banking protocols that allow blood and bone marrow collection as well as the collection of relevant clinical data. All patients who consent and have samples available are trialed for PDX development. In addition, samples from the Children’s Oncology Group (COG) are also trialed for PDX generation. Serially transplanting PDX models are validated using short tandem repeat (STR) and characterized using both targeted DNA/RNA next generation sequencing and RNAseq. As of March 2023, this systematic approach has resulted in 26 serially transplanting models. Models have been shared with requesting labs to facilitate external pAML pre-clinical studies. Available PDX models can be located through the BCM PDX Portal. We expect our growing PDX resource to make a significant contribution to expediting the testing of promising novel therapeutics for pAML.

## 1. Introduction 

Pediatric acute myeloid leukemia (pAML) accounts for about 18% of all childhood leukemias and is the second most common type of acute leukemia in children [1,2]. In the United States, roughly 730 patients under the age of 20 are diagnosed with pAML each year [3]. Adult AML is more common than pAML, affecting patients with a median age of 65. In the US, adult AML has an incidence rate of 1.8:100,000 persons for those under 65 and 17:100,000 for those over 65 [3]. As with pAML, adult AML is also characterized by a lower survivability when compared to acute lymphoblastic leukemia [4]. Although similarities exist between adult and pediatric AML, there are notable differences in their genetic and biological makeup. For example, a structural variant, most often a fusion, typically drives pAML as opposed to adult AML [5]. 

Since pAML differs from adult AML biologically and genetically, it is crucial to develop agents specific to pAML. More than five novel agents have been approved for use in AML in the past 4 years [6]. Midostaurin, gilteritinib, and CPX-351 are among the novel agents that have recently been proven effective against AML. Other new agents like ivosidenib and enasidenib have demonstrated promising results in treating AML with specific genetic mutations [7,8]. While some agents developed for adult AML may be useful in pAML, many of the mutations that these agents target in adult AML are rare in pAML, so they are unlikely to benefit pAML patients. For example, while IDH1 and IDH2 inhibitors have shown great promise in adults with AML who harbor these mutations, these mutations are present in less than 1% of children with pAML; these new agents are therefore unlikely to improve outcomes for 99%+ of pAML patients. This example highlights the importance of directed drug development for pAML.

Patient-derived xenograft (PDX) models are the gold standard for pre-clinical novel agent evaluation [9,10]. These models are created by transplanting human cancer cells into immunodeficient mice (IDM), allowing for the faithful replication of the human tumor in a live animal model [9,11]. PDX models have been successful in pre-clinical drug testing, accurately reflecting the genetic and biological heterogeneity of the human disease and its response to treatment [12]. Many cancer subtypes have large collections of PDX models available for pre-clinical testing [13]. These collections allow novel agents to be tested in multiple different genetic subtypes to understand ranges of responses [9]. Historically, however, developing PDX models from AML patients has been difficult. Improved IDM models have better engraftment rates. Two IDM mouse strains, NSG (NOD-Scid-IL-2Rγcnull) and NSGS (NOD-scid IL2Rgnull-3/GM/SF), are commonly used to generate PDX models. The NSGS strain allows for better and faster engraftment and in many cases, the presence of a leukemic burden similar to that seen in patients’ peripheral blood, which facilitates simple blood sampling for longitudinal monitoring. However, one potential downside of the NSGS strain is that leukemia progression can sometimes be quite rapid, which makes using the models for pre-clinical studies difficult [14]. MISTRG (M-CSF^h/h^ IL-3/GM-CSF^h/h^ SIRPa^h/h^ TPO^h/h^ RAG2^−/−^ IL2Rg^−/−^) mice are another promising IDM strain that lends itself to AML engraftment by providing a supportive microenvironment for the development and maintenance of hematopoietic stem and progenitor cells (HSPC). However, this strain is not commercially available at this time, and engraftment rates still remain relatively low. Despite these hurdles, many labs have focused on developing novel therapies and PDX models for adult AML, and with more patients and more labs working on developing adult AML PDX models, many such models now exist [15].

In contrast, few pAML PDX models exist for several reasons, including limited patient numbers, the lack of specialized techniques and expertise required to generate models, and a shortage of financial incentive to develop models for a relatively rare disease [9]. A handful of researchers have successfully developed some pAML PDX models [16,17,18,19,20,21]. Nevertheless, the scarcity of pAML PDX models hinders the development and prioritization of new treatments for pediatric AML patients and leaves a gap in the pipeline for new therapies. Thus, generating pAML PDX models is essential for improving treatment options for children with AML. Texas Children’s Hospital at Baylor College of Medicine (TCH/BCM) has had recent success in addressing this gap in the pre-clinical pipeline that evaluates novel agents. The team at TCH has generated multiple pAML PDX models. This success has been facilitated by a high volume of pAML patients, existing specimen banking protocols, and efficient coordination between the clinic and labs. Here, we present our collection and the protocols we have established to generate and validate pAML PDX models. Viably banked cells from pAML PDX models established in commercially available mice are available to the research community at no cost. With this breakthrough, TCH wants to supply institutions with clinically relevant pAML PDX models to speed the evaluation of novel agents to target pAML.

## 2. Materials and Methods 

### 2.1. Patient Samples

Patient samples for the study were obtained from two sources: locally banked samples from children diagnosed at TCH and banked samples from the Children’s Oncology Group (COG). Both sources included samples from male and female children, with the majority of COG samples coming from children enrolled on the AAML1031 phase 3 trial for de novo AML (NCT01371981) (Figure 1). The trial enrolled children and young adults up to age 30, but our collection only includes samples from patients who were under age 21 at diagnosis. 

AAML1031 had a male to female ratio of 55% to 45% and the racial/ethnic distribution was 79% white, 14% black/African American, and 7% Asian. Fifteen percent of the patients were Hispanic or Latino; the remaining 85% were not Hispanic or Latino. Meanwhile, the patient demographics at TCH are enriched for Hispanic ethnicity, with 45% of the population being Hispanic or Latino and 55% non-Hispanic or Latino. TCH patients were enrolled on at least one of two local tissue banking protocols, and both trials were open to patients of all sex, gender, racial, and ethnic backgrounds. These protocols were approved by the Baylor College of Medicine Institutional Review Board. We obtained written informed consent (and assent as appropriate, BCM IRB#24170 initial approval 12/16/2008, IRB#3342 initial approval 28 March 2001) for the use of patient samples from patients and/or parents/legal guardians per institutional, local, and federal policies and in accordance with the Declaration of Helsinki.

### 2.2. Mice

The program used NSG and NSGS mice obtained from Jackson Laboratories (JAX stock #005557 and #013062) and subsequently bred in-house; new breeding mice were replaced with commercial supply biannually [22,23]. Both of these mice strains are immunodeficient animals that lack functional B and T lymphocytes accompanied with multifunctional defects in natural killer (NK) cellular activity. NSGS mice also express human GM-CSF, IL-3, and stem cell factor driven by a CMV promotor. The NSGS strain has been shown to enhance the engraftment of human cells of the myeloid lineage [23]. The immunodeficient mouse strains MISTRG (VG5097/5090/5155/5089/5079/5078) and MISTRG6 (VG5097/5090/5155/5089/5079/5078/790) express human cytokines (M-CSF, IL-3/GM-CSF, and TPO), have a human SIRPα knock-in to the mouse SIRPα locus, and were developed by Regeneron Pharmaceuticals and the Flavell laboratory [24,25]. We were able to use these mice to establish pediatric AML-derived xenografts through a materials transfer agreement with Regeneron Pharmaceuticals. For attempting pAML PDX generation, the use of NSGS mice was prioritized due to the ease of working with these commercially available mice. When a patient sample was known to harbor a high-need mutation or genetic anomaly, the sample was also trialed for engraftment in MISTRG or MISTRG6 if mice were available. For all strains, both male and female mice 5–8 weeks of age were used for PDX engraftment attempts. Mice were housed in a private immunodeficient suite within the BCM Center for Comparative Medicine at TCH. Murine work was approved as described by the BCM IACUC (initial approval date 27 December 2011).

### 2.3. Tail Vein Injection (TVI) of Samples 

For initial attempts to establish a PDX, 2 × 10^5^ cells/mouse of isolated mononuclear cells (MNCs) from patient bone marrow, peripheral blood, or pheresis material were injected by tail vein at 10^5^ cells/100 µL of saline. In preparation for TVI, mice were restrained in a TVI injection platform (Braintree) and lateral caudal tail veins were dilated with a heat lamp. Tuberculin syringes (27 gauge) were used for sample injection. In rare cases when only minimal patient sample was available (100–200 µL starting volume), the sample was red cell lysed but not enriched for MNCs and then resuspended in saline for TVI. For serial passages, 2 × 10^5^ AML blast cells isolated from the bone marrow (BM) or spleen from previous passages were utilized for TVI. Tissues with the highest percentage of AML blast cells were utilized, typically >85%.

### 2.4. Peripheral Blood (PB) Monitoring for Engraftment

Starting at 4 weeks after injection, PB samples were collected and analyzed using flow cytometry for human AML cells. If the mice looked clinically ill before the four weeks had elapsed, then engraftment checks began earlier. Peripheral blood was collected for testing every two weeks or more often if the mouse appeared ill. Peripheral blood was collected from the lateral caudal tail vein after restraint in a TVI mouse holder, and a scalpel or cutting edge needle was used to nick one of the lateral caudal tail veins. Approximately 50–100 μL of PB was collected in a capillary tube, transferred to an EDTA tube, and processed for flow cytometry (according to the methodology outlined in the Section 2.7).

### 2.5. Euthanasia and Harvesting Tissue

Once human AML cells reached at least 40% in peripheral blood, or if mice were clinically ill, the mice were humanely euthanized and tissues were harvested. Mice were euthanized through isoflurane inhalation followed by cervical dislocation to ensure death prior to tissue collection. Prior to tissue collection, mice were sprayed with isopropyl alcohol to saturate all hair and skin. The chest wall was opened and peripheral blood was collected via right ventricle puncture under direct visualization. During harvest, the spleen and femurs were routinely collected. For mice harboring pAML samples that are expected to be fibrotic, the tibias and spine were also collected. A single cell suspension was made from the spleen in Iscove’s Modified Dulbecco’s Medium (IMDM) using a gentleMACS (Miltenyi Biotec, North Rhine-Westphalia, Germany). Long bones (femurs/tibias) were cut at one end to expose the bone marrow, and the bones were spun down in an Eppendorf (1 min at 2500× *g* at 25 °C) to collect bone marrow cells before their resuspension in IMDM. All samples underwent a cell count so that their allocation to TVI/genomic testing/storage aliquots post-flow cytometry could be arranged. Aliquots of the spleen, bone marrow, and peripheral blood samples were evaluated using flow cytometry to quantify the disease burden in each tissue as shown for a representative sample in Figure 2 (following the methodology outlined in the Section 2.7). If engraftment had occurred, the spleen and BM samples were cryopreserved, and an aliquot of cells was resuspended in saline for serial transplantation into the next cohort of mice (via tail vein injection). We harvested some mice’s sternums to evaluate bone marrow histology.

### 2.6. Serial Transplantation of Samples

Cells from BM or spleen from mice that had engrafted were once again transplanted into new mice via TVI at a concentration of 2 × 10^5^ AML cells in 200 µL of saline per mouse (as outlined in the Section 2.3). Every serial transplantation cycle involved a minimum of three new mice injected with cells from the harvested mice. We monitored secondary and subsequent passages of serially transplanting samples the same way we did for primary passages, described above, but began peripheral blood engraftment checks 2–3 weeks before this engraftment was first documented during the first passage. We recorded subsequent passages’ time to engraftment and banked MNCs isolated from harvested tissues for future use and distribution.

### 2.7. Flow Cytometry

Once obtained, peripheral blood, bone marrow, and spleen samples from mice were processed and analyzed with flow cytometry to assess the degree of engraftment. As previously described, peripheral blood and spleen samples were collected and then suspended in 1 mL of red blood cell lysis buffer (500 mL distilled water, 4.4 g NH_4_Cl, 0.6 g Tris) and rocked for 15–20 min. The PB and spleen samples were moved from their EDTA tubes to 1.5 mL micro centrifuge tubes and centrifuged at 400× *g* for 5 min at 4 °C; supernatant was aspirated from the cell pellet. After cell preparation, all samples (PB, spleen, and BM) and the positive control cell line were stained with fluorochrome conjugated antibodies for human CD45-APC-H7, CD33-AF700, and CD3-PE (BD Biosciences or Biolegend) and passed through a 40µm nylon cell strainer (Corning) prior to data collection on an LSRII or Symphony (BD Biosciences). Distinct hCD33+/hCD45+ cell populations indicated engraftment. Human CD3 positivity indicated T cell rather than pAML cell engraftment. For PB engraftment checks, we considered engraftment achieved when hCD45+/hCD33+ in PB was ≥1%. Mice with ≥40% of human AML cells in PB were considered to have met criteria to be humanely euthanized and harvested.

For the extended flow cytometry panel, the banked PDX or corresponding diagnostic AML cryopreserved cells were quickly thawed in a 37 °C water bath, and then transferred into a 15 mL conical tube. Flow wash solution (1XPBS/2%FCS/0.01%NaN_3_) was added dropwise while gently vortexing, and then centrifuged at 350 G for five minutes. The supernatant was decanted, and cells were resuspended in 2 mL of flow wash solution. A total of 100 µL of resuspended cells was added to the tubes containing titered cocktailed antibodies from the ΔN:M™ test [26], then vortexed and incubated at room temperature in the dark for 20 min. After incubation, 1 mL of flow wash solution was added to each tube, vortexed, and centrifuged at 350 G for five minutes. The supernatant was decanted and the cells were resuspended in flow fixative solution (1% formalin) then analyzed on a FACS Calibur (BD Biosciences, San Jose, CA, USA). Each channel of the flow cytometer was standardized to yield fluorescence intensities in terms of numbers of antigenic molecules per cell (not MFI). The quantitative antigen expression, comparing the diagnostic specimen to the corresponding PDX models, is shown in Appendix A.

### 2.8. Tissue Banking

Select tissues (BM and spleen) from mice with engrafted pAML patient samples were banked. MNCs were isolated through Ficoll density separation using lymphocyte separation media (Corning, Somerville, MA, USA). Both BM and Spleen MNCs were resuspended after isolation in freezing media (80% Fetal Bovine Serum, 10% RPMI, and 10% DMSO) at 1.5 × 10^7^ cells/mL for BM and up to 3 × 10^7^ cells/mL for spleen MNCs.

### 2.9. DNA Extraction/Short Tandem Repeat (STR) Testing

For patient samples that engrafted at least twice and were considered serially transplantable models, STRs were done from tissue collected after the first passage to validate the model. DNA was extracted from MNCs from the source patient sample and from MNCs harvested from mice after the first passage of the sample using the QIAmp DNA mini kit. Paired samples underwent STR testing of 13 autosomal loci to validate the PDX model’s identity.

### 2.10. Histology

Sternums from mice harboring pAML PDX samples were first fixed and decalcified in Cal-Ex II overnight, and then processed in a Tissue-Tek VIP using routine methods. The samples were cut at 4 um’s and H&E was performed in a Gemini AS. Patient BM core biopsies had been previously processed in the same way, and representative images were selected for comparison.

### 2.11. Fluorescence In Situ Hybridization (FISH)

FISH was performed to validate KMT2A rearrangements in AML001, AML005, and AML018, and a gain of 1q25 region in AML004. The FISH probes (KMT2A and 1p36/1q25 Dual Color Probes) were obtained from Abbott, Abbott Park, IL. Hybridization was performed according to the manufacturer’s protocols. The slides were counterstained with 4,6-diamidino-2-phenylindole (DAPI), and the images were captured using 10×/25 aperture at a magnification of 100×/1.40 oil R on a Nikon E800 microscope equipped with a cooled charge-coupled device (CCD) camera. The cells were analyzed using Smart capture 3 (Digital Scientific UK, Cambridge, UK). A total of 200 interphase nuclei were analyzed for rearrangement/copy number changes.

### 2.12. Targeted Next-Generation Sequencing

Targeted next-generation DNA sequencing (TCH-HDP) was performed on extracted DNA from paired diagnostic and PDX (*n* = 20) samples to detect mutations in the *TERT* promoter and coding and splicing regions of 3126 exons in 174 genes (Appendix A), as described previously [27]. Copy number variant analysis was performed as previously described [28].

Variants, copy number alterations, and gene fusions were categorized as ‘Tier I/II’ (established or potential clinical significance) or ‘Tier III’ (unknown clinical significance), based on evidence of actionability and clinical significance, following professional guidelines and curated cancer resources [29,30].

## 3. Results 

### 3.1. Description of Patient Samples Used to Generate PDXs

To better understand the characteristics of pAML samples that developed into stable PDX models, the samples were analyzed for their ability to create serially transplantable models. A total of 174 samples were collected from 154 patients (from Texas Children’s Hospital and from the Children’s Oncology Group). For samples from the Children’s Oncology Group, only one sample per patient was tested. For samples from Texas Children’s Hospital, at times we tested multiple tissue types (i.e., PB and BM for patients with a challenging BM aspirate and scant supply for local research), and if a sample was available at the time of additional clinical events (i.e., induction failure or relapse), those samples were also trialed. Of the 174 samples trialed, 49 resulted in a primary engraftment and 26 succeeded in transplanting serially (Figure 3A). More than 50% of the 26 serially transplanting samples had a *KMT2A* rearrangement, despite this driving mutation being much less frequent in the general pAML population (Figure 3B, Table 1) [5]. Other alterations represented in our serially transplanting models included gene fusions involving *CBFA2T3*, *NUP98*, and *PICALM,* and a *RUNX1* mutation. Notably, patient samples with core binding factor pAML, inv(16) and t(8;21), or *CEBPA* did not produce even primary engraftment. For patients with *KMT2A* rearrangement, 29% had a sample engraft, while the rate was 14% for *NPM1* and 20% for both *FLT3* and *CBFA2T3::GLIS2*. Regarding the cell type used to establish models, MNCs from BM was the most common cell source, peripheral blood was the second most common, and pheresis product the third. Cells from pheresis product were slightly more likely to produce a stable PDX model, possibly related to the enrichment of these samples for *KMT2A* rearrangements. (Figure 3C). In our real world attempts to engraft patient samples into immunodeficient mice, we found that samples from Texas Children’s Hospital, whether fresh or viably preserved, were slightly more likely to engraft a primary passage when compared to samples from the Children’s Oncology Group (34% vs. 20.8%). However, for both groups, about half of samples that engrafted once engrafted a second time (52% and 56%, respectively). This may have been in part due to fresh samples having somewhat superior initial engraftment rates when compared to viably preserved samples (32% vs. 25%). Again, once the first passage was engrafted, about half of the samples engrafted a second time.

### 3.2. Time to Engraftment and Strains Engrafted

The serially transplanting models were all characterized for disease burden at engraftment, the length of time to engraftment, and time to moribund. Amongst the 26 models described here, all models developed measurable PB AML and could be monitored by hCD45 and hCD33. The time for models to become moribund ranged from 5 to 28 weeks, with a mean of 12.8 weeks. For some models, patient samples were trialed in multiple different strains and PDX models engrafted in other strains. Notably, models AML007 and AML008 only engrafted in MISTRG or MISTRG6 strains while the remainder engrafted in NSGS. These models harbor *PICALM::MLLT10* and *RAD21* mutations, respectively. As of March 2023, all models passaged at least twice and as many as 6 times, with a mean of 3.5.

### 3.3. Characterization of Stable PDX Models

All models described here have been validated using STRs. Additionally, several models have been compared for differences in histology between source patient bone marrow biopsy material and bone marrow histology from mice harboring the same samples. Across models for which histology was performed, bone marrow at the time of organ harvest demonstrated the replacement of normal hematopoiesis with a near-complete infiltration of pAML blasts (Figure 4A). Most mice also developed an infiltration of leukemia blasts within the spleen, which provided an easily accessible source of cells for tissue banking. Other than splenic disease, extramedullary disease was rarely identified in mice (with the exception of AML011, which developed extramedullary masses, one of which was adjacent to the sternum with histology as shown (Figure 4A)). FISH was also completed for a subset of models demonstrating marker fidelity in all models tested (*n* = 4, Figure 4B). Most importantly, given frequent concerns about subclone loss after the passaging of patient samples, DNA next-generation sequencing panel testing revealed that pAML PDX models retained similar variant allele frequencies when compared to their source patient material (Figure 4C). Of the 26 serially transplanting models described here, 20 had NGS done, with 19 of the 20 having paired patient and PDX sequencing completed (15 shown in Figure 4C and 4 samples with no Tier I/II variants). One sample had PDX-only testing completed (AML901). Of the 15 samples for which we had paired patient/PDX sequencing with Tier I/II variants, 11 samples (73.3%) were concordant for all variants; samples AML904, AML906, and AML019 harbored some but not all patient variants, and 1 sample out of 20 was discordant (6.7%; AML014; the patient’s *KRAS* variant was not detected in the PDX model). Flow cytometry comparisons between paired diagnostic patient material and MNCs from pAML PDX models performed at Hematologics demonstrated some mild variability in surface and intracellular marker expression between the patient and PDX samples. The extended flow cytometry panel examining 11 cell surface markers was completed on six of the serially transplanting models presented. Two additional surface markers were examined for three of the paired samples. The markers were not identical between the diagnostic specimen and PDX cells but appeared to be similar in many aspects. Between the myeloid and lymphoid markers, the myeloid markers seemed to be more representative of the diagnostic specimens whereas the lymphoid antigens had the most change after transplantation. Similar to the myeloid markers, the progenitor markers were shown to be durable and representative of the diagnostic specimen. The fractions of lymphoid antigens that remained similar between the diagnostic and PDX samples were 5/6 for HLA-DR, 4/6 for CD38, 1/6 for CD7, and 2/6 for CD56. The fractions of myeloid antigens that remained similar were 2/6 for CD11b, 4/6 for CD36, 5/6 for CD13, 5/6 for CD33, and 5/6 for CD14. The progenitor antigens CD34 and CD117 were similar between diagnostic and PDX samples in 4/6 and 6/6, respectively. We hypothesize that changes in cell surface phenotype may be related to the effects of the murine microenvironment on the pAML cells. Our work highlights the importance of testing markers of interest prior to utilizing PDX models for experiments targeting cell surface antigens (Appendix A).

### 3.4. Public Availability of Models

The Pediatric Acute Leukemia Xenograft (PALeX) collection utilizes the BCM PDX Portal to disseminate information about publicly available models. The BCM PDX Portal includes information on the source patient for each model, time to engraftment, disease characteristics, and key genetic information on each model. When available, the Portal also has detailed information on the treatment the source patient received, response to each treatment trialed, and clinical outcomes. PALeX models are updated within the Portal at least biannually and can be accessed at https://pdxportal.research.bcm.edu/pdxportal, (accessed on 11 November 2023).

## 4. Discussion 

Overall survival rates for pAML vary depending on several factors, including genetic mutations and response to treatment. Compared with adult AML, in which the goal is often disease control, the goal for pAML is cure. Adult AML and pAML are biologically distinct; this results in many of the novel agents that show promise in adult AML not being useful for a large subset of pAML [5]. Prioritizing the development of novel agents and translation to the clinic has been hampered in pAML by some gaps in the pre-clinical pipeline, one of which is the need for robust and genetically diverse pAML PDX models. 

Patient-derived xenograft (PDX) models are created by implanting fragments of tumor tissue (or malignant cells for liquid tumors) obtained directly from cancer patients into immunodeficient mice. This allows for the creation of models that closely resemble the original patient’s tumor and can be used to study tumor characteristics and response to treatments. PDX models have been shown to be more predictive of patient responses to therapy than traditional cell line models, as they typically retain the heterogeneity and genetic complexity of the original tumor. Multiple studies have demonstrated the utility of PDX models in pre-clinical drug development for cancer, with published studies demonstrating a correlation between PDX and patient clinical response as early as the 1980s [31,32]. While genetically engineered mouse models have the benefit of being able to evaluate interactions between AML cells and the immune microenvironment, PDX models remain the current pre-clinical gold standard for testing novel agents in a biologically relevant manner. Compared to genetically engineered models, PDX models better represent the intratumor heterogeneity that can contribute to treatment resistance. Humanized mice offer the potential for both the engraftment of a more diverse panel of PDX models and for the ability to study immune interactions with malignant cells and novel therapeutics, but these are not widely available and are a challenge to work with. Three-dimensional ex vivo models with AML cell lines and support cells have been described but are less well studied for patient samples [33]. Thus, PDX models remain a crucial tool for the pre-clinical testing of novel agents.

Recent studies have utilized PDX models to identify new therapeutic targets and test new drugs for pediatric acute lymphoblastic leukemia (pALL). One illustrative study used PDX models to demonstrate the efficacy of the BCL-2 inhibitor venetoclax in treating a subset of pALL cases [34]. Because of the ease of generating pALL PDX models, 40 pALL PDX models were generated from 36 pALL patients, and these models retained the original malignancy’s molecular and genetic characteristics. Subsequently, drug testing using the PDX models demonstrated single-agent and combination activity in response to venetoclax as well as several other tested drugs. This experience highlights one of the potential ways to utilize PDX models for rare diseases to evaluate and prioritize potential new therapeutics.

While PDX models have become an important pre-clinical tool to study malignancies, developing PDX models can be a difficult and time-consuming process, and particularly challenging for myeloid-derived malignancies. Our collection has several models with KMT2A fusions, a subtype of AML has been described to engraft relatively well in immunodeficient mice, and other models with more rare pAML genetic findings. Notably, among the 26 serially transplanting pAML PDX models described here, there were two, AML007 and AML008, that only serially transplanted in the MISTRG strain. The first model was derived from a patient with a PICALM::MLLT10 fusion, ETV6 loss, and a NRAS mutation. The second model came from a patient with a RAD21 mutation. In our real-world experience, attempting to engraft all available patient samples into immunodeficient mice, we found that many genetic subtypes of pAML do not engraft well in a primary passage and that many do not serially transplant well either. While the explanation for this is unknown, it likely has to do with supportive conditions within the host microenvironment that allows pAML to become established and expand. Work is ongoing to optimize the engraftment of rare subtypes of pAML, including host conditioning and the use of alternative immunodeficient mouse strains. Despite the challenges related to their generation, PDX models have shown great promise for developing novel agents for AML, especially for older adults. In contrast, drug development for pediatric AML has been relatively stagnant for the past two decades. One of the only new approved treatment options is gemtuzumab ozogamicin, a CD33-targeting antibody–drug conjugate used in combination with standard chemotherapy.

Our systematic approach to developing pAML PDX models has been successful thus far and is providing a renewable resource for pAML researchers at TCH/BCM and elsewhere. It is our hope that establishing this publicly available resource will speed the identification, testing, and selection of the most promising new targets and drugs for pAML.

## 5. Conclusions

PDX models provide a valuable pre-clinical tool for studying the biology and treatment of pAML. The PALeX Program at Texas Children’s Hospital has successfully created numerous validated pAML PDX models using samples acquired from consenting patients. We have been able to serially passage, characterize, bank, and distribute these newly developed models to other institutions for their own research. This helps address the gap in the pre-clinical pipeline for testing of promising new agents and mechanisms of chemotherapy resistance. Additionally, with a constant influx of new pAML patients and samples, we will continue to generate new models. For those interested in obtaining pAML PDX models from our lab, please visit (https://pdxportal.research.bcm.edu/pdxportal, (accessed on 11 November 2023)) for more information. By retaining the biological and genetic characteristics of the original patient tumors, these PDX models can provide new insights into the pathogenesis of AML and may lead to the development of new, more effective therapies.

## Figures and Tables

**Figure 1 biomedicines-12-00394-f001:**
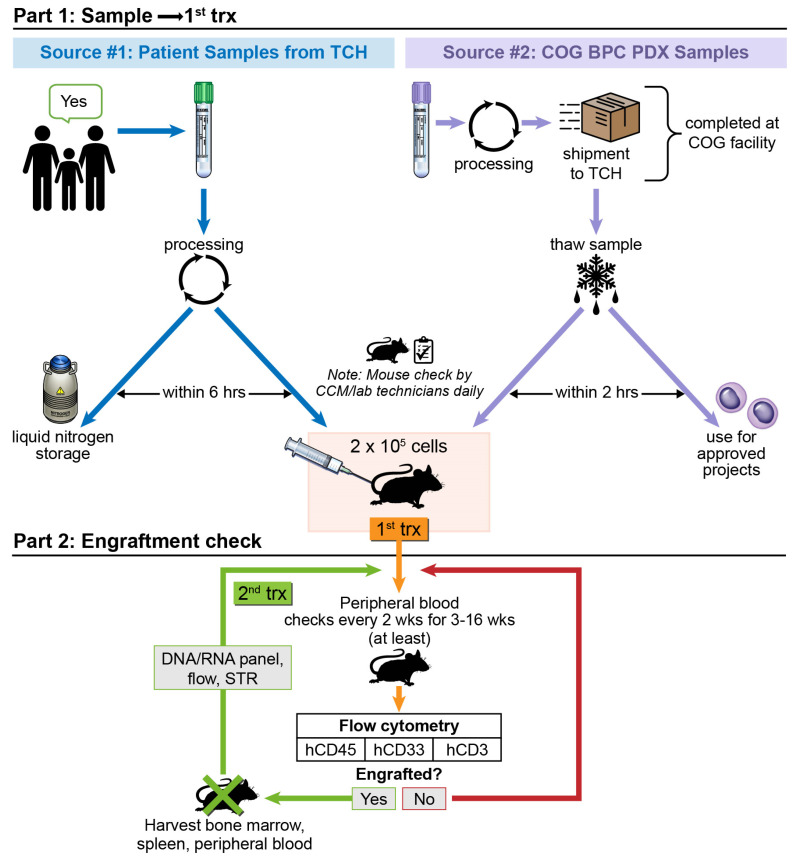
Schematic depicting workflow for PDX model establishment. Patient sample sources are either from TCH patients or from patients who have consented to COG tissue banking studies. Abbreviations: Trx: transplant; TCH: Texas Children’s Hospital; COG: Children’s Oncology Group; BPC: Biospecimen Processing Center; STR: short tandem repeat.

**Figure 2 biomedicines-12-00394-f002:**
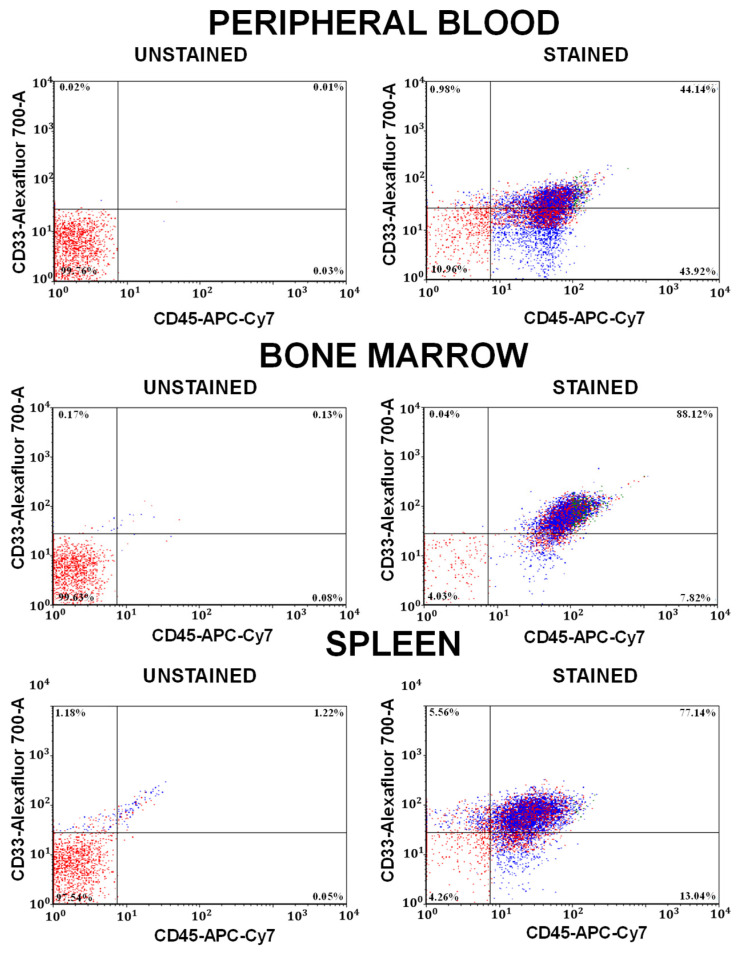
Representative engraftment plots. Flow cytometry dot plots of the engraftment of pediatric AML in the peripheral blood, bone marrow, and spleen of a pAML PDX model. Representative sample AML005a; the third transplant from NSGS is shown. Note: some auto fluorescence is seen in the unstained spleen sample.

**Figure 3 biomedicines-12-00394-f003:**
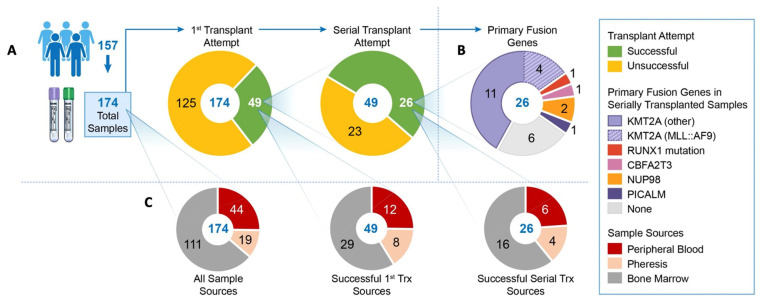
Samples used to generate PDXs. (**A**) The pie charts depict the number of consenting participants, the number of samples collected from patients, the proportion of samples that produced a primary PDX, and the final number of serially transplantable samples. (**B**) The pie chart shows the primary fusions or mutation of the 26 patient samples that produced serially transplanting PDX models. (**C**) The pie charts designate the site of sample collection for all 174 samples, the 49 samples that succeeded in producing a primary PDX, and the 26 that produced serially transplantable PDX models.

**Figure 4 biomedicines-12-00394-f004:**
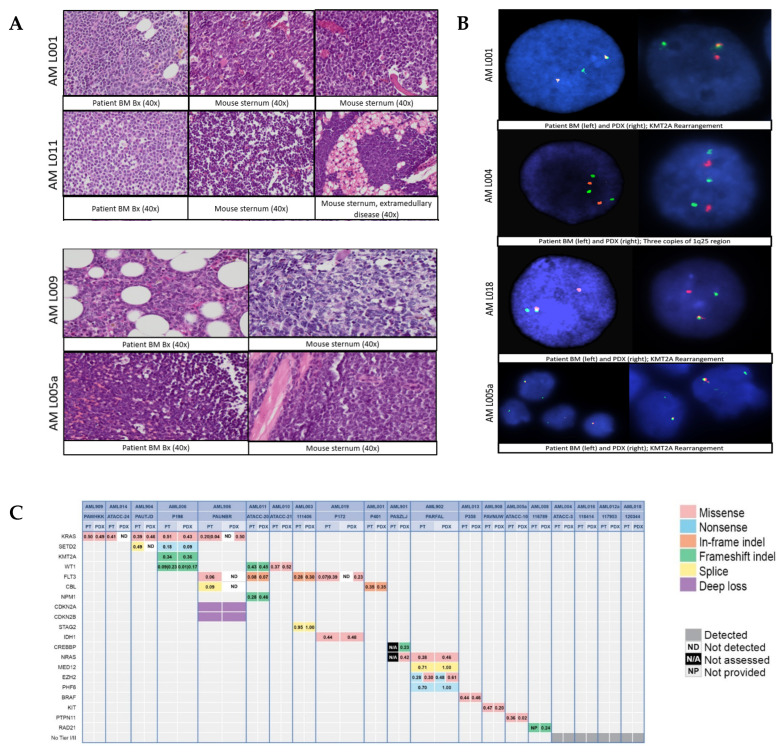
Characterization of established PDX models. (**A**) Histology between source patient bone marrow biopsy and murine bone marrow is similar as shown in four representative cases. (**B**) FISH assessment for recurrent mutations demonstrates the presence of key fusions in PDX model and source patient material in four representative cases with the red probe indicating the 3′ *KMT2A* gene and green the 5′ *KMT2A* gene. (**C**) Next-generation sequencing of patient vs. PDX DNA demonstrates the fidelity of models and the preservation of variant allele frequencies in most cases. Models 4, 12a, 16, and 18 did not have Tier I/II DNA mutations identified and are not included in this graphic. Of note, the PDX model AML008 had a RAD21 mutation that was identified on clinical patient testing but variant allele frequency was not provided on clinical testing from the patient.

**Table 1 biomedicines-12-00394-t001:** Characteristics of 26 of the models in the TCH pAML PDX collection—patient age at diagnosis (years), race and ethnicity, key mutations, the number of successful passages through mice, time until moribund, and immunodeficient mouse model strains engrafted are shown. Viably frozen cells from pAML PDX models established in commercially available mice are available to the research community at no cost. Definitions: M: male; F: female; W: white; B: black; A: Asian; AI: American Indian; H: Hispanic; NH: non-Hispanic; Dx: diagnosis; t-AML: therapy-related AML; VUS: variants of uncertain significance; N: NSG; M: MISTRG; and M6: MISTRG6.

Public Model Name	Age/Sex	Race/Ethnicity	Timepoint	Major Cytogenetic or Molecular Features	Passages	Time to Moribund—NSGS Weeks, Mode (Range)	Other Strains Engrafted
AML001	1/M	W/NH	Dx	ins(10;11)	5	7 (7–24)	N, M, M6
AML002	20/F	B/NH	Relapse	t(7;21), −17	4	17 (9–18)	
AML003	7/M	W/H	Dx t-AML	*FLT3* (2 PMs), *FANCG*, *STAG2*	2	10 (9–13)	
AML004	1/M	W/NH	Dx	t(1;21), *IDH2*, *MUTYH*	5	9 (6–13)	M6
AML005a	16/M	W/H	Dx	3′ KMT2A deletion/KMT2A::MLLT4, PTPN11	2	10 (8–18)	M, M6
AML006	9/M	B/NH	Relapse	*KMT2A::MLLT1*, *KRAS*, *SETD2*, *WT1*	4	8 (7–9)	N, M, M6
AML007	11/M	W/NH	Induction Failure	*PICALM::MLLT10*, *ETV6 loss*, *NRAS*	2	N/A	M6
AML008	4/F	W/NH	Dx	*RAD21*	2	N/A	M, M6
AML009	19/F	W/H	Relapse	*KMT2A::ENL*, *TP53*	3	16 (11–16)	M6
AML010	2/M	W/NH	Dx	+10/*WT1*	3	5 (5–6)	M, M6
AML011	19/F	W/H	Dx	*FLT3-ITD*, *NPM1*, *WT1*	3	26 (16–46)	M6
AML012a	15/M	W/H	Dx	*NUP98::NSD1*, *FLT3-ITD*, *WT1*	2	35 (13–35)	M
AML013	13/M	B/NH	Dx	t(6;11)/*KMT2A::MLLT4*	2	8 (8–15)	M, M6
AML014	0.6/F	W/NH	Dx	*KMT2A::MLLT3*	2	16 (12–35)	M6
AML016	15/F	W/H	Dx t-AML	multiple VUS	5	18 (12–26)	
AML018	3/M	B/NH	Relapse	*KMT2A::MLLT3*, *ABL1*, *PTCH1*	6	5 (4–11)	M6
AML019	11/M	W/NH	Dx	*FLT3* (2 PMs), *IDH1*	2	27 (11–29)	
AML901	10/F	W/NH	Dx	*KMT2A::MLLT10*, −10	4	10 (9–11)	
AML902	19/M	W/NH	Dx	*PICALM::MLLT10*	5	18 (11–18)	
AML903	17/F	Other/NH	Dx	*NPM1*	4	11 (11–11)	
AML904	2/M	W/NH	Dx	*KMT2A::MLLT10*	4	10, 12 (10–17)	
AML905	1/F	W/H	Dx	*CBFA2T3::GLIS2*, +8, +21	4	9 (8–15)	M6
AML906	15/M	A/NH	Dx	*KMT2A::MLLT3*	5	7 (6–16)	
AML907	15/M	W/NH	Dx	*NPM1*	6	16 (15–42)	
AML908	10/F	B/NH	Dx	*KMT2A::MLLT3*	4	12 (11–26)	
AML909	0.5/F	AI/H	Dx	*KMT2A::AFF3*	2	10 (8–18)	

## Data Availability

The data that support the findings of this study are available from the corresponding author upon reasonable request.

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
