# Peer review of "Addressing a Pre-Clinical Pipeline Gap: Development of the Pediatric Acute Myeloid Leukemia Patient-Derived Xenograft Program at Texas Children’s Hospital at Baylor College of Medicine"

_biomedicines, 2024, doi:10.3390/biomedicines12020394_

Round 1

Reviewer 1 Report

Comments and Suggestions for Authors

This paper identifies and seek to address the current need for additional relevant pediatric AML models for use in a variety of applications including preclinical drug testing. Overall, the manuscript sufficiently describes the rationale for generating pAML PDXs and methodology behind model generation and validation. However, there are a few items that require additional comment and that would strengthen this paper.

1) The authors describe generating a total of 26 models that were amenable to serial propagation from a total of 174 samples received from Texas Children's and the COG repository (~15% overall establishment rate). It would be useful to know what the breakdown in samples received separately from TCH and COG. Were there differences in degree of engraftment and serial establishment from samples obtained from the two sources.

2) Were there models that were established using "fresh" sample or were all samples implanted all derived from viably cryopreserved material? If there was a mixture of samples implanted as fresh and viably cryopreserved samples, were there observed differences in rates of engraftment and serial propagation?

3) The authors describe 2 models that only engrafted in MISTRG or MISTRG6 strains. Where there molecular features that would explain for why these models engrafted only in MISTRG and not in NSGS?

4) It is unclear if all models were validated in terms of confirming recapitulation of the mutational signature from matched source patient-PDX pairs. If this was not done for all models, what proportion of models had this analysis performed and what was the degree of patient-PDX concordance (and discordance)?

5) Similar to comment #4, to what degree does the immunophenotype of matched pt-PDX pairs recapitulated? What were the concordance (and discordance) rates?

6) Given an overall engraftment rate of ~28% and establishment/propagatable rate of ~15%, can the authors comment on factors that may have led to relatively low engraftment/establishment rates? Is this due to biologic features or processing of material or both? Are there suggestions or work to improve these rates?

7) There has been other efforts to generate pAML models including work by Scott Armstrong and this should be reference in Introduction and/or Discussion. 

Reviewer 2 Report

Comments and Suggestions for Authors

The manuscript by Stevens et al. provides relevant hints on the generation of PDX from pediatric AML, which represents a research advancement for the potential application to the therapeutic testing. In particular, the deposition of publicly available models is highly appreciable.

Some important points need to be addressed. In particular, while the methods section is very detailed, some results sections remain vague:

- the number of passages performed in serial transplantation experiments needs to be reported

- which stage of disease was used for the generation of PDX? Was it the disease diagnosis? The higher number of samples used compared with the number of patients is due to the usage of diverse tissues or multiple time points?

- 3.1: the percentage of successful engraftment also in relationship with the diverse genetic backgrounds need to be reported in details and then discussed in the Discussion.

- 3.2: diverse murine strains were used for the generation of PDX. The reason that guided these choices need to be described along with the differences obtained in terms of percentage of engraftments also in relationship with the genetic background. Can the authors provide any indications for the strains to be selected? Please also discuss that.

- AML007 and AML008 only engrafted in MISTRG or MISTRG6. Please describe the patients’ genetic alterations and discuss.

- The complete Table of the genetic alterations pf engrafted patients need to be reported, conversely Figure 5 can be removed from the manuscript.

- 3.3: please discuss potential reasons of the diverse surface phenotype between cells from the patients’ and the PDX.

- Table 1: what does Time to Moribund - NSGS (weeks) refer to? Is it the first transplantation? Is it a disease acceleration through serial passages? Please describe and comment.

- The discussion section should contextualize the advantages but also the limitations of the PDX models, including the lack of or the limited interactions with the immune microenvironment. Moreover,  it should compare the PDX models with novel frontiers as humanized mice and 3D ex vivo model of leukemia.

Minor points:

- please unify the two methodologies sections on flow cytometry

Round 2

Reviewer 1 Report

Comments and Suggestions for Authors

Revisions and responses appropriate and adequate to proceed with acceptance in current form. No additional comments or suggestions.